# Work Stressors and Intention to Leave the Current Workplace and Profession: The Mediating Role of Negative Affect at Work

**DOI:** 10.3390/ijerph192113992

**Published:** 2022-10-27

**Authors:** Arkadiusz M. Jasiński, Romuald Derbis

**Affiliations:** Department of General and Work Psychology, Institute of Psychology, University of Opole, 45-040 Opole, Poland

**Keywords:** work-related stress, turnover intention, affect at work, stressor-emotion theory, midwives

## Abstract

The first aim of this study was to determine whether organisational constraints, interpersonal conflicts at work, workload and negative affect at work have a positive relationship with intention to leave a current job or profession. The second aim was to investigate whether negative affect at work mediates the relationship between work stressors and intention to leave a current job or profession. The study was a quantitative cross-sectional design in nature. Data were collected between March and April 2022. The sample consisted of 306 midwives working in the Polish public health service. The theoretical model was tested using structural equation modelling. The results confirmed positive direct relationships between workload and negative affect with intention to leave the current workplace and intention to leave the midwifery profession in general. The relationships between organisational constraints and interpersonal conflicts at work and intention to leave a job or profession were found to be completely mediated by negative affect at work. Our study revealed that workload is the strongest direct predictor of intention to leave the current job or profession. Organisational constraints and interpersonal conflicts at work lead to an intention to leave a job or profession by inducing negative affect at work. Interpersonal conflicts at work are the strongest predictor of negative affect at work.

## 1. Introduction

The most sought-after profession in the European Union (EU) requiring tertiary education is a healthcare profession [1]. In Poland, too, there is a growing problem of accessibility to health care services, including those with a gynaecology-obstetrics and neonatology profile. The number of places available in gynaecology-obstetrics wards is steadily decreasing, from 20,293 places in 2004, to 17,806 in 2018, and 15,060 in 2020 [2]. Similarly, in neonatology wards, 9106 beds were available in 2010, decreasing to 7967 in 2020. The number of midwives is also decreasing, and the age structure of the profession does not guarantee generational replacement. As many as 55% of Polish midwives are between 51 and 70 years of age [3] The cited source indicates that midwives may incur a significant health cost from their work, as the average age of death in this profession is 61.5 years, compared to the average age of women in the general population of 81.8 years. The EU healthcare sector is the most exposed to emotionally exhausting social behaviour at work [4]. The statistics cited above directed our attention to the issue of the potential relationship between stressors and negative emotions experienced at work with the intention to leave the current workplace or profession.

Our article is the first empirical study on this topic in the Polish population. The study was conducted in Poland but addresses a broader problem, namely women’s access to gynaecological and obstetric care. Our research is important because it identifies possible reasons for midwives to leave their job or profession. The deficiency of qualified midwives in Poland is a fact, and the forecasts suggest that this could worsen. The average age of a midwife in Poland is 51 years and 25% of nurses and midwives working in Poland have retirement rights and may leave the profession at any time [3]. Declining staffing levels may lead to a lower quality of care and, in the long term, may expose women to exclusion in health care. Given these staff shortages, one of the public policy priorities at national and international level should be to reduce turnover in the midwifery profession. In our opinion, one of the key factors responsible for the intention to leave a job or profession are stressors and negative emotions in the workplace. The aim of the research presented here was to examine, based on stressor-emotion theory [5], whether and how organisational constraints at work, workload, interpersonal conflict, and negative affect at work are associated with intention to leave the current workplace or the profession.

## 2. Stressors at Work and Intention to Leave Current Job or Profession

We define intention to leave as the attitude to leave the current workplace (e.g., hospital, clinic, ward, dispensary) or to give up the midwifery profession altogether. According to us, attitude is a source of motivation, so it can result in the performance of a specific action. Intention to leave is not equivalent to actually leaving the organisation, but intention has a high predictive value of making such a decision [6]. It is therefore highly likely that midwives who declare an intention to leave are thinking intensely about their decision to change their current job or midwifery profession. Previous research shows that the heavy nature of a job [7], the mismatch between job expectations and working conditions [8], and job stress [9,10] are positively associated with intention to leave that job or profession. The vast majority of studies to date have posited the determinants of intentions to leave a job or profession in a group of midwives as low job satisfaction [11,12]. According to the stressors-emotions model, negative behaviours towards work and the organisation, including resignation from work and the profession, may be the result of long-term experience of stressors at work and failure to cope with them [13]. Our research explains what role stressors and negative affect at work play in shaping this intention.

Stressors at work are physical, organisational, social, and psychological factors that generate high work costs. Stressors engage an employee’s energy, and coping with them requires long-term effort, which can lead to negative consequences for the employee and the organisation. One of the negative effects of experiencing excessive stressors in one’s work may be the decision to change jobs or change careers. In our article, we use the definition of stressors proposed by Spector and Jex [14]. They distinguish three general stressors at work: organisational constraints, workload, and interpersonal conflicts at work. They are the most frequently indicated qualitative stressors at work by employees [15]. Previously, these stressors have been studied, among others, in the context of their associations with social support, work engagement, and job burnout [16,17], but not for the midwifery group. Below, we substantiate the role of the identified three work stressors as predictors of intentions to leave a job or the profession.

Organisational constraints are phenomena in the workplace that prevent employees from transforming competence and effort into high professional performance [14]. According to the cited authors, work constraints include poor equipment, lack of resources, unclear procedures and responsibilities, interruption of work, and lack of or inadequate support from other employees. Previous research has shown that organisational constraints are negatively related to motivation and job performance [18]. In addition, organisational constraints are positively related to low job satisfaction, low job commitment, and the intention to change or leave a job [19]. A relationship between job burnout and intention to leave the current workplace has also been demonstrated [20]. Based on Spector and Jex’s stressor theory discussed above, and the results of the cited studies, we hypothesised the following:

**H1a.** 
*Organisational constraints at work are positively related to the intention to leave the current workplace.*


**H1b.** 
*Organisational constraints at work are positively related to intention to leave the profession.*


The second stressor we studied is workload. In our study, workload has a quantitative dimension; that is: the degree of workload is measured by the amount of work that employees have to do in a certain period of time. Thus, the perceived workload will be the result of experiencing the pressure to perform quickly at work, to exert a great deal of effort at work, and to perform an amount of work that subjectively exceeds the employee’s resources. This understanding of workload derives from the definition formulated by Spector and Jex and from empirical data. The results of a study among Australian midwives suggest that workload is mainly due to time demands [21]. The study revealed that time pressure occurs for 42% of working days in this group and is positively associated with frustration, depression, and reduced work activity. Chronic strain is experienced by up to 79.5% of midwives [22]. Previous research has shown that workload is negatively related to motivation and job performance [18], and is a predictor of lower job satisfaction, higher stress [23], and intention to leave the occupation [24]. If the midwives studied experience high workload (i.e., among other things, they have to act quickly and their daily work requires a lot of physical and mental effort), then it is highly likely that experiencing this stressor at work may be positively associated with considering leaving the job or profession. The results of the literature review led us to hypothesise:

**H2a.** 
*Workload is positively related to the intention to leave the current workplace.*


**H2b.** 
*Workload is positively related to intention to leave the profession.*


Interpersonal conflict at work refers to how often an employee experiences conflict, such as an argument with others or colleagues who are unkind to them. These phenomena are accompanied by aggression and hostility, which may occur incidentally or chronically. They may manifest as gossiping about colleagues, rude behaviour, verbal conflicts, serious disputes or physical violence [14]. Interpersonal conflicts at work result more from negative attitudes towards colleagues than from individual differences [25]. Medical professionals indicate that interpersonal conflicts at their work are most often caused by a misunderstanding of job roles and the responsibilities assigned to them [26]. Other results suggest that interpersonal conflicts at work are positively associated with emotional exhaustion and absenteeism, especially with low social support at work [27]. Interpersonal conflict between employees is a positive predictor of stress and negative attitudes towards work [28]. Higher levels of interpersonal conflict at work lead to lower intrinsic motivation [29] and increased frustration, anger, anxiety, depressive symptoms, and burnout in employees [30]. The previous studies proved the positive relationship between interpersonal conflicts and turnover intention [31,32]. However, those cited research were not focused on the healthcare sector. Based on the above considerations, we hypothesise:

**H3a.** 
*Interpersonal conflicts at work are positively related to the intention to leave the current workplace.*


**H3b.** 
*Interpersonal conflicts at work are positively related to intention to leave the profession.*


### Negative Affect at Work as a Mediator between the Work Stressors and Intention to Leave

We believe that the intention to leave a job or profession is a psychological reaction to negative aspects of the organisation or job. This reaction has an emotional, cognitive, and behavioural component. The emotional component describes emotions and feelings towards the job, whereas the cognitive component includes what the employee thinks about their job, and the behavioural component is a visible act of will (an action motivated by the other two factors). We hypothesise that the long-term experience of stressors at work will be positively associated with negative affect at work and with the intention to change jobs or occupations.

Stressors can cause negative affect at work because negative emotional states are an immediate reaction to a stressful situation [13]. We define affect at work as relatively enduring consciously available feelings [33], which are employees’ emotional reactions to their work. For negative affect, unpleasant emotional tension is typical [34], which, by inducing avoidance motivation, may determine the positive relationship between negative affect and the intention to leave the job or profession. Experiencing negative affect at work is associated with feelings of a ‘negative sign’ and high (scornful, nervous, jittery) or low (sluggish, sleepy, bored) agitation [35]. Negative affect at work, understood as a subjective emotional appraisal of work, can be influenced by the level of stressors at work. Earlier studies have confirmed the positive association of organisational constraints, workload, and interpersonal conflict with negative affect [36,37,38]. In concluding these considerations, we hypothesise:

**H4a.** 
*Organisational constraints are positively related to negative affect at work.*


**H4b.** 
*Workload is positively related to negative affect at work.*


**H4c.** 
*Interpersonal conflict at work is positively related to negative affect at work.*


Previous studies have shown that the intention to leave the current workplace or profession can be a way of coping with stressors and the negative emotions that accompany them [39]. It can therefore be assumed that midwives experiencing stressors at work will experience more negative affect and this will lead to an intention to leave the job or profession. To date, the mediating role of negative affect in the relationship between stressors and negative attitudes towards work, mainly counterproductive behaviour, has been repeatedly demonstrated [40,41]. However, the mediating role of negative affect in the stressors-intensity relationship among midwives was not considered. We therefore further hypothesised:

**H5a.** 
*Negative affect at work is positively related to the intention to leave the current workplace.*


**H5b.** 
*Negative affect at work is positively related to intention to leave the profession.*


**H6a.** 
*Negative affect at work mediates the relationship between work stressors and the intention to leave the current workplace.*


**H6b.** 
*Negative affect at work mediates the relationship between work stressors and intention to quit.*


The conceptual model and hypotheses are presented in Figure 1.

## 3. Materials and Methods

### 3.1. Aims

Our study had two objectives. The first was to determine whether organisational constraints, workload, interpersonal conflicts, and negative affect at work are positively related to the intention to leave one’s current job or profession. The second aim was to test whether negative affect at work mediates the relationship between work stressors and intention to leave the job or profession.

### 3.2. Design

The study was cross-sectional in nature. The sample size needed for the study was calculated using G*Power v. 3.1.9.2 [42]. With an assumed mean effect size (*f*^2^) of 0.15 and a significance level of less than 0.001, the minimum quantity required was 300 observations. The final sample comprised 306 midwives, indicating a reliability sample power of 99.50%.

### 3.3. Participants

The study sample consisted of 306 midwives employed full-time in the Polish public health service. The midwives worked in south-western Poland. All study participants were women.

### 3.4. Data Collection

Participants were recruited in-person by the researcher at their place of work or study. Respondents were assured that the research was voluntary and anonymous. Participation in the study was free of charge and participants could withdraw at any time. Only midwives employed full-time and working directly with patients were eligible for the study. Some of the respondents were on midwifery master’s programmes. Data were collected between March and April 2022. The study consisted of completing a battery of psychological questionnaires and took an average of 15 min to complete. Only complete questionnaires were included in the analyses. In order to reduce bias in self-report measures, respondents were only informed about the general purpose of the study, without indicating the dependent variable and its postulated relationships with the explanatory variables. In order to reduce common method error, spatial spacing was used [43]; that is: questionnaires that measured the key variables within the study were separated from questionnaires that were not the focus of the study. In order to increase the validity of the postulated causal relationships, the order in which the variables were measured followed the order in which they were placed in the empirical model under test.

### 3.5. Ethical Considerations

The research design was reviewed and approved by the relevant university ethics committee. Following approval from the university, formal permission to conduct the study was sought from facilities employing or training midwives. Once the relevant approvals were obtained, forms were made available through volunteer midwives and in-person. Each participant gave informed written consent to participate in the study.

### 3.6. Measures

Sociodemographic data were collected using a self-administered questionnaire. The questionnaire included questions on sex, age, marital status, length of service, branch of work, number of working hours per week, number of jobs, region of work, and education.

Intention to leave a job was measured using the question: “Are you very seriously considering leaving your current job?”. Intention to leave the profession was measured using the question: “Are you very seriously considering leaving the midwifery profession?”. For each item, respondents answered on a scale from 1 (not considering at all), to 10 (considering every day). The literature indicates that the reliability of single-item measures is as high as multi-item measures, provided they are used to measure concrete rather than abstract and complex constructs [44]. Concrete constructs are those that, among other things, theoretically consist of a single component, are not the sum of assessments of several aspects (such as job satisfaction and customer satisfaction), and do not include factors such as personality traits, extraversion (which can manifest as sociability), risk-taking, and activity [45]. Intention to leave the current job or profession is, in our opinion, a focused and semantically unambiguous variable [46] that can be reliably measured with a single-item instrument. In our study, we wanted to find determinants of general intention to leave a job or profession, in which case a one-item scale is adequate [47]. The other variables in the study were measured with multi-item scales because, due to their complex structure, the single-item recommendation cannot be generalised beyond concentrate constructs. The use of single-item and short multi-item measures was also intended to keep survey time to a minimum to prevent the possible perception that participation in the study was an additional psychological burden.

Work stressors were measured using three scales developed by Spector and Jex [14], adapted to Polish [30]: Organisational Constrains Scale, Quantitative Workload Inventory, and Interpersonal Conflict at Work Scale.

The Organisational Constrains Scale is used to measure work-related constraints such as those in the sphere of resources, equipment, policies, and procedures. It has a single-factor structure, consisting of 11 responses to the question “How often do you find it difficult or impossible to do your job because of…?” with options “Poor equipment or supplies” and “Organizational rules and procedures”. Answers are given on a five-point Likert scale, ranging from 1 (*less than once a month or never*) to 5 (*several times a day*). The level of organisational constraints is inferred from the sum of all responses. The range of possible scores is between 11 and 55. In our study, the reliability was Cronbach’s α = 0.86.

The Quantitative Workload Inventory is used to measure quantitative workload expressed in terms of the effort required to perform a large number of tasks within a limited time. It consists of five questions measuring one factor, such as: “How often are you forced to do more work than you are able to?” Answers are given on a five-degree Likert scale ranging from 1 (*less than once a month or never*) to 5 (*several times a day*). The range of scores is between 5 and 25. In our study, the reliability of the scale was Cronbach’s α = 0.85.

The Interpersonal Conflict at Work Scale measures the frequency of experiencing disputes or conflicts and rude behaviour in the workplace. One factor is measured with four questions, such as: “How often do you get into arguments with others at work?”. Answers are given on a five-point Likert scale, ranging from 1 (*less than once a month or never*) to 5 (*several times a day*). The range of scores is between 4 and 20. In our study, the reliability was Cronbach’s α = 0.82.

Negative affect at work was measured using the Job Affect Scale [48] in a Polish adaptation [49]. The technique allows for the measurement of positive and negative affect experienced at work. In this study, only the scale measuring negative affect was used. It is estimated using 10 terms that best reflect negative affect, which, in factor analyses, correlate most strongly with the factor interpreted as negative affect [50]. They denote unpleasant involvement and dislike of life and action (e.g., scornful, nervous, jittery, sluggish, sleepy, bored). For each term, answers are given on a seven-point scale, ranging from 1 (*very weakly*) to 7 (*very strongly*) with respect to the intensity of feeling a given affective state at work in the past two weeks. The range of possible scores is between 10 and 70. The reliability of the scale in the presented study was Cronbach’s α = 0.79.

### 3.7. Data Analysis

The dependent variables were intention to leave the current workplace and intention to leave the profession. The research model was constructed to explain the relationships between stressors and negative affect at work and intention to leave a job or profession, and whether the relationship between stressors and intention to leave a job or profession was mediated by negative affect at work. First, the distributions of the variables were checked using the Shapiro–Wilk test, and descriptive statistics and Pearson correlation (r) coefficients were calculated. Each scale was analysed separately and then the whole research model was checked using Confirmatory Factor Analysis (CFA). The maximum likelihood estimation method was used. Goodness of fit indices were used to assess the goodness of fit of the model: χ^2^/df, the goodness of fit index (GFI), the comparative fit index (CFI), the Tucker–Lewis index (TLI), incremental fit index (IFI), standardised root mean square residual (SRMR), and the root mean square error of approximation (RMSEA). The χ^2^/df index should be less than three [51]. With GFI, CFI, TLI, and IFI values above 0.95, and RMSEA below 0.06, model fit is good, and GFI, CFI, TLI, and IFI values above 0.90 and RMSEA below 0.08 are adequate [52]. SRMR should be less than 0.08 [53]. The research model was tested using structural equation modelling (SEM). We tested the hypothesised direct and mediation effects simultaneously, because testing for effects individually may falsify results and lead to misinterpretations [54]. Data were processed using SPSS 22 and Amos.

### 3.8. Validity, Reliability, and Rigour

Reliable Polish adaptations of open-access tools, used repeatedly in previous studies, were used in the research presented here. All items of the scales that were used significantly loaded onto general factors at a level above 0.79. The scales achieved a sufficient level of reliability, with Cronbach’s α ranging from 0.79 to 0.86.

## 4. Results

Subjects ranged in age from 23 to 61 years (*M* = 34.00, *SD* = 12.03). The mean length of service was 11.12 years (*SD* = 11.81, range 1 to 39 years). The midwives surveyed were characterised by low scores on the scale of intention to leave the current workplace (*M* = 3.60, *SD* = 2.68) and to leave the profession (*M* = 2.05, *SD* = 1.98). The Pearson correlation (r) coefficient between age and tenure was *r* = 0.97, *p* < 0.001. Sociodemographic data are presented in Table 1. Analysis of intergroup differences in sociodemographic variables showed no statistically significant differences between midwives in the level of psychological variables studied. The absence of collinearity between predictors was confirmed. The VIF collinearity coefficient was less than 1.37 for each of the explanatory variables and the tolerance index ranged from 0.73 to 0.91. Descriptive statistics and correlation coefficients are presented in Table 2.

### 4.1. Confirmatory Factor Analysis (CFA) Results

The results of the CFA showed that all items of the questionnaires used were statistically significantly related to their main factors at a level above 0.79 (*p* < 0.001). Model fit indices indicated a good fit of the final model to the data χ^2^/df = 1.34, GFI = 0.926, CFI = 0.970, TLI = 0.964, IFI = 0.971, SRMR = 0.0526, RMSEA = 0.034. We also tested whether theoretical distinctiveness of the constructs tested occurs at the empirical level. We tested a model composed of four separate predictors (four-factor model) and an alternative model composed of a single factor containing all scales (one-factor model). The results showed that the four-factor model had an acceptable fit, confirming the theoretical distinctiveness of the constructs tested. Acceptable CFA results allowed the hypotheses to be tested.

### 4.2. Direct Effects Analysis Results

The results of the path analysis indicate that organisational constraints at work were not statistically significantly directly related to intention to leave the current workplace (*β* = −0.09, *SE* = 0.10, 95% CI [−0.306, 0.110], *p* = 0.237) or to intention to leave the profession (*β* = −0.03, *SE* = 0.09, 95% CI [−0.245, 0.112], *p* = 0.615). On this basis, Hypotheses H1a and H1b were rejected. Workload was a positive direct predictor of both intention to leave the current job (*β* = 0.19, *SE* = 0.06, 95% CI [0.057, 0.328], *p* = 0.003) and intention to leave the occupation (*β* = 0.13, *SE* = 0.06, 95% CI [0.018, 0.246], *p* = 0.039). On this basis, Hypotheses H2a to H2b were accepted. Interpersonal conflicts at work were not positively significantly directly related to intention to leave the current workplace (*β* = −0.06, *SE* = 0.15, 95% CI [−0.452, 0.123], *p* = 0.393) or intention to leave the occupation (*β* = −0.006, *SE* = 0.14, 95% CI [−0.396, 0.164], *p* = 0.944). Hypotheses H3a and H3b were rejected. Organisational constraints (*β* = 0.22, *SE* = 0.09, 95% CI [0.040, 0.414], *p* = 0.015) and interpersonal conflicts at work (*β* = 0.34, *SE* = 0.13, 95% CI [0.093, 0.627], *p* < 0.001) were positive predictors of negative affect at work. Workload was not positively related to negative affect at work (*β* = 0.08, *SE* = 0.09, 95% CI [−0.110, 0.266], *p* = 0.272). Hypotheses H4a and H4c were therefore accepted, while hypothesis H4b was rejected. Negative affect at work was positively related to intention to leave the current workplace (*β* = 0.44, *SE* = 0.18, 95% CI [0.237, 0.941], *p* < 0.001) and intention to leave the occupation (*β* = 0.40, *SE* = 0.17, 95% CI [0.217, 0.911], *p* < 0.001). This confirms Hypotheses H5a and H5b.

### 4.3. Indirect Effects Analysis Results

The results of the mediation analysis indicate that the relationship between organisational constraints and intention to leave the current workplace was completely mediated by negative affect at work: the standardised total effect (X→Y) was *β* = 0.009, *SE* = 0.08, 95% CI [−0.162, 0.178], *p* = 0.927, and the standardised indirect effect was *β* = 0.09, *SE* = 0.06, 95% CI [0.024, 0.288], *p* = 0.011. Negative affect at work completely mediated the relationship between interpersonal conflict at work and intention to leave the workplace: total effect (*β* = 0.08, *SE* = 0.07, 95% CI [−0.066, 0.240], *p* = 0.279, indirect effect (*β* = 0.15, *SE* = 0.14, 95% CI [0.034, 0.650], *p* = 0.012). Negative affect at work did not mediate the relationship between workload and intention to leave the workplace: indirect effect (*β* = 0.03, *SE* = 0.05, 95% CI [−0.055, 0.154], *p* = 0.339). Hypothesis H6a was therefore partially accepted. The association between organisational constraints and intention to leave the occupation was completely mediated by negative affect at work: total effect (*β* = 0.052, *SE* = 0.06, 95% CI = −0.079, 0.190], *p* = 0.455), indirect effect (*β* = 0.09, *SE* = 0.06, 95% CI [0.021, 0.279], *p* = 0.005). There was full mediation by negative affect at work between interpersonal conflicts at work and intention to leave: total effect (*β* = 0.13, *SE* = 0.07, 95% CI [−0.010, 0.278], *p* = 0.074), indirect effect (*β* = 0.14, *SE* = 0.13, 95% CI [0.035, 0.605], *p* = 0.005). Negative affect did not mediate the relationship between workload and intention to quit, such that the indirect effect was *β* = 0.03, *SE* = 0.04, 95% CI [−0.051, 0.146], *p* = 0.336. Hypothesis H6b was therefore partially confirmed. The developed model helped to explain 28% of the variation in negative affect, 21% of the variation in intention to leave the current workplace, and 18% of the variation in intention to leave the profession. The results of the path analysis are included in Figure 2.

## 5. Discussion

The aim of this study was to test whether work stressors and negative affect at work are positive predictors of intention to leave the current job or profession. In addition, we tested whether the relationship between work stressors and intention to leave a job or profession is mediated by negative affect at work. The results of this study indicate that workload leads to both types of intention to leave directly. Organisational constraints and interpersonal conflicts at work increase the intention to leave a job or profession indirectly, by inducing negative affect at work.

The midwives surveyed had higher levels of intention to change jobs than professions. We can suppose that Polish midwives have similar reasons of intention to leave their job/profession to Dutch [55], British [56], Australian [57], and Canadian [58] midwives. However, we want to emphasize that research methods and databases were different, so that our results and those cited cannot be reliably compared. Mentioned previous studies do not involve the mediating role of negative affect at work. So, the presented results have some novelty. Namely, in addition to the important role of work stressors, we indicated the emotional foundations of intention to leave the current workplace and profession. The observed relatively low intentions to leave the current workplace or profession of Polish midwives may be due to socio-economic variables, such as the desire to recoup the costs invested in specialised education, the small number of alternatives to practising midwifery in a location other than an organised, publicly funded facility (independent midwives are not popular in Poland), and high job security and income stability (i.e., more jobs than midwives). Perhaps the lower intention to leave observed in this study is due to the fact that, in the Polish health service, the intensity of social stressors at work is lower than the European average and twice as low than in countries with the highest intensity of such behaviour in Europe: Netherlands, Denmark, France [4]. Moreover, the low turnover intention can be determined by the midwives’ conviction of the mission to help others. Woman-centred care is the defining feature of midwifery [59]. Unlike in previous studies, age, job tenure [11], education level [60], and marital status [61] were not significantly associated with the intention to leave a job or occupation. The discrepancies indicated suggest that sociodemographic variables are not a universal determinant of functioning at work.

The purpose of testing hypothesis group H1 to H3 was to test whether organisational constraints, workload, and interpersonal conflicts at work lead directly to increased intention to leave the current workplace or profession. Findings suggest that, of the stressors studied, only workload is in a direct positive relationship with intention to leave the job and intention to leave the profession. This result is in line with previous studies, in which the compulsion to perform a large amount of work in a short period of time, and working overtime, were positively related to the intention to leave the organisation and to leave the profession prematurely [62]. When controlling negative affect at work, organisational constraints and interpersonal conflicts at work are not directly related to intention to leave the job or profession. Previous studies have demonstrated direct relationships between job burnout [63], general job stress [64], specific interpersonal and organisational stressors, and intention to leave the workplace [10]. Our findings add to the existing body of knowledge insofar as they suggest that the relationship between stressors and intention to leave the workplace or profession is more complex. The lack of a direct relationship between organisational constraints and interpersonal conflicts at work with intention to leave a job or profession is probably due to the fact that these stressors do not lead to negative attitudes towards work directly, but instead indirectly by inducing a specific psychological mechanism (e.g., a mechanism based on negative emotions). We describe this mechanism below when discussing the results on the indirect effects hypotheses.

In Hypotheses H4a to H4c, we hypothesised that stressors would lead to increased negative affect at work. The current results do not support previous suggestions of a positive direct relationship between workload and negative affect [37]. In the cited studies, the relationships between work stressors and negative affect were tested separately; whereas, in our study, those relationships were tested in a single statistical model. The lack of a significant workload–negative-affect relationship may be due to controlling for the other two stressors. In such a model, they appeared to be more important in explaining negative affect at work than workload. The positive relationship we observed between organisational constraints and interpersonal conflict at work and negative affect is consistent with previous results [36,38]. Our study supports theoretical conjectures about the emotional effects of experiencing stressors at work. Negative affect at work can be triggered by organisational constraints and interpersonal conflicts at work because it is an immediate adaptive bodily response to stressors: demands exceeding the individual’s ability to cope [13]. Our findings suggest that midwives experience negative affect at work because they need to respond to working conditions in some way. If their work is fraught with stressors, the natural emotional response to aversive environmental stimuli may be negative. Interpersonal conflicts at work are stronger predictors of negative affect compared to organisational constraints. This is a complementary result to previous studies finding a positive association between interpersonal conflicts and negative emotions [65], although our results suggest that social stressors are more strongly associated with negative affect than organisational stressors.

Hypotheses H5a to H5b addressed the positive relationship between negative affect at work and intention to leave a job or profession. As in previous studies, we found that the intention to leave a job or profession may be a result of experiencing negative emotional states at work [39]. Negative affect is more strongly associated with the intention to leave the current workplace than with the intention to leave the profession. Our study, although partly in line with previous studies, contributes several new findings to the literature. The preponderance of previous research on the consequences of stress in midwives’ work did not explicitly address intention to leave, but explained the determinants of burnout [66] or identified resources and demands in their work [67]. We have shown that stressors directly (workload) or indirectly through the arousal of negative affect can not only lead to counterproductive behaviour [41], but can also result in an intention to leave the job or profession.

In Hypotheses H6a to H6b, we suggested that negative affect mediates the relationship between stressors and both types of intention to leave. The results confirmed our conjecture. It appears that the intention to leave one’s current job or profession may be a behavioural response to negative affect induced by work stressors. Based on our results, it can be said that midwives experiencing organisational constraints and interpersonal conflicts at work are characterised by a higher intensity of negative affect at work and, consequently, a higher intention to change jobs or professions. Our findings are in line with previous suggestions treating intention to leave as a finality of experiencing excessive stress at work and not coping with it. Feijen-de Jong et al. [55] demonstrated that the decision to leave a job is the result of maladaptation to unfavourable working conditions, high work demands, and frustration experienced due to failure to cope. Our findings support broader theoretical assumptions about the consequences of affectively negative stimuli on motivation [68]. Negative work factors (e.g., organisational constraints and interpersonal conflict), by inducing negative affect, are positively related to motivation to resign (withdrawal) from a job or profession. In light of the indirect effects results, the above-mentioned relatively low level of intention to leave the job or profession of Polish midwives can also be explained in psychological terms. Previous studies indicate that job insecurity is positively associated with negative emotions at work [69]. If Polish midwives have a sense of job insecurity, they should thus experience negative feelings about work less often. We showed that a key determinant of the translation of stressors (organisational constraints and interpersonal conflicts at work) into intention to leave is the negative affect experienced in relation to work. Future research could test our hypothesis that the higher the job stability, the lower the negative affect and the lower the intention to leave the job or profession.

Our research was mainly conducted on the basis of the stressors-emotions model. The results presented here confirm the theoretical assumptions of this model for several reasons. Firstly, we found that organisational constraints and interpersonal conflicts at work were positively related to negative affect at work. Secondly, we demonstrated that organisational constraints and interpersonal conflicts at work led to intentions to leave a job or profession through inducing negative affect at work. Thus, we have confirmed that negative actions towards work and the workplace may be the result of experiencing long-term stressors at work and failing to cope with them [13]. Frustration in coping with stress leads to negative emotions towards work. Thirdly, we confirmed that midwives shape their behaviour based on affective states. The results presented here complement the broader context of research on the affective determinants of workplace behaviour and the consequences of emotional labour [70]. The mechanism we describe is that the behavioural response to stressors that are beyond midwives’ capacity (the intention to leave the current job or profession) is the result of a response to the negative affect induced by these two stressors, and not of chronic experiences of organisational constraints and interpersonal conflict at work. It is noteworthy that workload is positively associated with the intention to leave work and the profession directly. Therefore, it can be said that workload induces both intentions to leave in midwives independently of the negative affect that they experience at work. The further research can be focused on two points: (1) identifying the resources that can lower the experienced level of work stressors and, thus, decrease negative affect at work and intention to leave levels; (2) deepening the positive social, organizational, and psychological factors which can buffer the negative relationship between work-related stress and intention to leave. For instance, the motivating role of job satisfaction can be checked. These propositions are based on a previous internationally aimed study which, in general, confirmed that job satisfaction is a significant negative predictor of intention to leave a job or profession among midwives [12]. However, the cited study was concerned on direct and general relationships. The further investigations can be oriented on indirect psychological mechanisms which can regulate the links between work stressors, job resources, job satisfaction, and negative job attitudes (including turnover intention).

### Limitations

Our research has some limitations. The first is the use of self-report measures only and the data were cross-sectional. This reduces the possibilities of generalising the results and inferring causal relationships. The countermeasures used (spatial spacing during measurement, a theoretically sound order of measurement of variables) and a statistical technique (path analysis) reduce the magnitude of error to some extent. In future studies, it is worth using a cross-lagged technique or a longitudinal study. Another limitation is the use of single-item measures for the dependent variables. This approach may raise legitimate concerns for opponents of such methods, particularly if intention to leave a job or profession would theoretically be a multidimensional construct. The analysis of the literature indicates that this is the first study devoted to intention to leave in a group of Polish midwives. Given the nature of the midwives’ work and the fact that most of them completed the questionnaires during their breaks at work (on the ward), we did not want to overload them with long sets of tests. Furthermore, we sought to justify that the leaving intentions studied could be treated as compact constructs and thus reliably measurable with a single-item scale. However, it would be worthwhile for future research to include a broader treatment of intentions to leave a job or profession, in order to validate the results. Sample size may also be a limitation. The statistical methods used were intended to increase the reliability of the results and to justify the conclusions on the basis of a limited number of observations. On the other hand, the collected sample, although relatively small, is to our knowledge the largest group of Polish midwives collected to date.

## 6. Conclusions

The results presented here suggest that workload (directly) and organisational constraints and interpersonal conflicts at work (indirectly via negative affect at work) are positively related to intention to leave the current workplace and intention to leave the midwifery profession. Interpersonal conflicts are the stressor most strongly associated with negative affect at work. The current findings draw attention to the importance of stress prevention in the healthcare sector. We encourage the healthcare managers to pay more attention to the degree of occupational stressors in midwives. This research may have practical value insofar as, assuming that workload directly predicts the level of intention to leave the job or profession, it is worthwhile to apply countermeasures to reduce the level of the indicated stressor. A solution to the problem would be to increase the number of midwives, which seems to lie with health policy makers. With a greater number of staff, there is likely to be a fuller on-call workforce and less overstaffing, resulting in less work under time pressure. Workshops to improve social skills, such as interpersonal communication, could also be considered as a solution given that organisational constraints and interpersonal conflicts at work are related to the intention to leave by arousing negative affect. This could raise the level of resources in the workplace in the long term. Overall, with adequate social support, the strength of the relationship between stressors, negative feelings at work and, ultimately, the intention to change jobs or leave the profession may decrease. Our study addressed a crucial problem, that is, providing healthcare at an appropriate level. In our opinion the deficiency of midwives is strongly associated with more duties for a single midwife, and, in final, the worse quality of healthcare.

## Figures and Tables

**Figure 1 ijerph-19-13992-f001:**
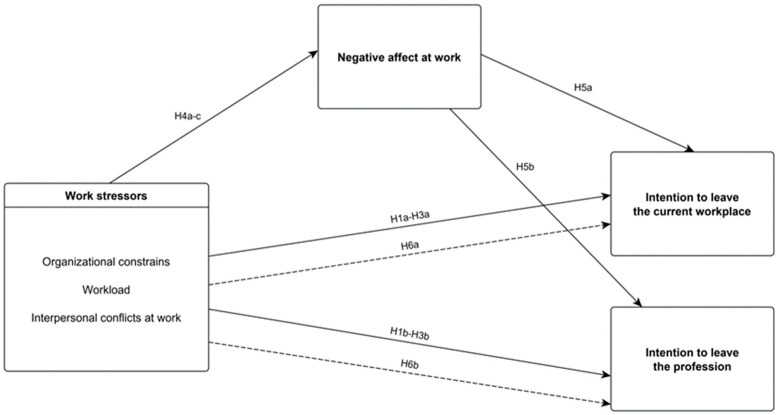
Research framework. Continuous lines indicate direct relationship between study variables. Discontinuous lines indicate relationship between work stressors and intention to leave mediated by negative affect at work.

**Figure 2 ijerph-19-13992-f002:**
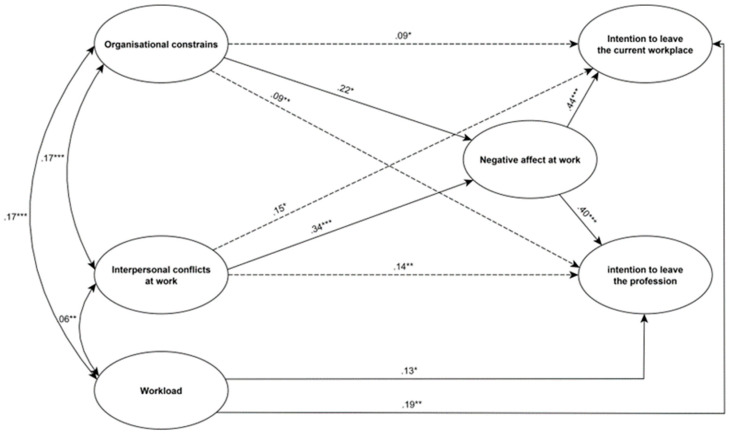
Standardized path coefficients of the final model (*N* = 306). Continuous lines indicate direct relationship between study variables. Discontinuous lines indicate relationship between work stressors and intention to leave mediated by negative affect at work. *** *p* < 0.001, ** *p* < 0.01, * *p* < 0.05.

**Table 1 ijerph-19-13992-t001:** Demographic characteristics of the study group (N = 306).

Demographics	Characteristics	Percentage (%)
Age [years] M (SD)	34.00 (12.03)	
Seniority [years] M (SD)	11.12 (11.81)	
Working time [h/week] M (SD)	43.55 (15.94)	
Work places	One	76.1
	Two	20.6
	More than two	3.3
Ward of working	Gyneacology-maternity	33
	Childbirth	19.2
	Gyneacology	10.7
	Maternity	9.6
	Neonatal	8.8
	Intensive therapy for newborns	5.4
	Pregnancy patology	4.6
	Clinic	4.2
	Obstetric and neonatal	4.2
	Own practice (one person)	0.3
Education	Midwifery high school	3.9
	Bachelor’s degree in midwifery	93.5
	Master’s degree in midwifery	2.6
Marital status	Single	50.3
	Married	43.7
	Divorced	2.8
	Non formal	2.8
	Widowed (one person)	0.3

**Table 2 ijerph-19-13992-t002:** Means, standard deviations, and correlations coefficients (Pearson’s r) between the study variables (N = 306).

Variable	*M*	*SD*	1	2	3	4	5
1. Organizational constrains	24.73	8.26	-				
2. Workload	19.41	4.48	0.27 ***	-			
3. Interpersonal conflicts at work	6.62	2.97	0.43 ***	0.18 **	-		
4. Negative affect at work	28.21	8.48	0.36 ***	0.18 **	0.33 ***	-	
5. Intention to leave the current workplace	3.6	2.68	0.12 *	0.22 ***	0.11 *	0.22 ***	
6. Intention to leave the profession	2.05	1.98	0.22 ***	0.19 **	0.18 **	0.25 ***	0.41 ***

*** *p* < 0.001, ** *p* < 0.01, * *p* < 0.05.

## Data Availability

Some or all data and models that support the findings of this study are available from the corresponding author upon reasonable request.

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
