# Peer review of "Work Stressors and Intention to Leave the Current Workplace and Profession: The Mediating Role of Negative Affect at Work"

_ijerph, 2022, doi:10.3390/ijerph192113992_

Round 1
Reviewer 1 Report
The topic of the analysed article is interesting. In order to be accepted for publication, a few aspects must be improved. At the end of the Introduction must be introduced a short description of the content of the article and insisting on the novelty and uniquess of the research. The literature review is well done, the sources are enough and some are recently published and from high ranked journals. The research is properly designed and follows logic steps in achiving the objectives. The research hypothesis are well determined and are accopmpanied by studies in the filed. The results are clear but the authors must improve the final part- the authors must add info about future research directions, and implications for management, for employees in the analysed field, for organizations, and also for society. Another point to improve or even change is reffering to the pfrase from lines 397-400: „The midwives surveyed had higher levels of intention to change jobs than professions. Polish midwives have a lower intention to leave their job/profession than Dutch (Feijen-de Jong et al., 2022), British (Hunter et al., 2019), Australian (Harvie et al., 2019), and Canadian (Stoll & Gallagher, 2019) midwives.”. This phrase must be changed, because the database for research is different, so the results cannot be compared. So, it is imposed to bring into discussion the studies and their results, but your results being different (maybe even the methods used into research are different), a comparison cannot be made. The authors is indicated to describe a little bit more those studies and their results. So, the article to be accepted for publishing, it is offered „Accept after minor revision (corrections to minor methodological errors and text editing)”.
Author Response
Dear Reviewer!
Thank you for your effort to improve our manuscript. We added all of your requirements. We hope our manuscript will be recommended for publication. All changes in the article are marked in red. We desribed our changes in atteched file.
Sincerely,
The Authors

Reviewer 2 Report
Overall, this is an interesting, clear, concise, well written and structured manuscript. The introduction is relevant and theory based. Sufficient information about the previous study findings is presented for readers to follow the present study rationale and procedures. The methods are generally appropriate. In sum, I find the manuscript is a high quality manuscript that covers thoroughly all relevant parts as a research paper. Once some editing issues and questions proposed by other reviewers that may have are addressed, I recommend this article to be published.
For more comments, please see the attachment.

Author Response
Dear Reviewer!
Thank you for your effort to improve our manuscript. We added all of your requirements. We hope our manuscript will be recommended for publication. All changes in the aricle are marked in red. We desribed our changes in atteched file.
Sincerely,
The Authors

Reviewer 3 Report
I thank the authors for an interesting view on Work stressors and intention to leave the current workplace 2 and profession: the mediating role of negative affect at work The article provides results that could be of interest to the professional public. Based on a systematic review, I would recommend revising this document for the following reasons: It is necessary to justify the research. Why is this important? Is your statistical sample representative? The literature used is outdated, I recommend expanding it with more recent sources Despite the quality of the article, I would welcome the expansion of the article in the area of ​​motivation. Many employees remain at their workplace despite unsatisfactory working conditions, which is especially true in the healthcare sector. It is from the conviction of the mission of these employees to help others. This was especially visible in recent years during the COVID-19 pandemic. You can get inspired by similar posts:
DOI: 10.3390/jrfm14100459
DOI: 10.17512/pjms.2021.24.1.04
DOI: 10.1080/1331677X.2021.1902365
DOI: 10.3846/tede.2020.13758
Author Response

(The authors gave the same response as above.)
